# Picking on the Same Person: Does Algorithmic Monoculture lead to Outcome Homogenization?

**Rishi Bommasani**[*]
Computer Science
Stanford University
nlprishi@stanford.edu

**Kathleen A. Creel**
Philosophy, Computer Science
Northeastern University
k.creel@northeastern.edu

**Ananya Kumar**
Computer Science
Stanford University
ananya@cs.stanford.edu

**Dan Jurafsky**
Linguistics, Computer Science
Stanford University
jurafsky@stanford.edu

**Percy Liang**
Computer Science
Stanford University
pliang@cs.stanford.edu

## Abstract

As the scope of machine learning broadens, we observe a recurring theme of *algorithmic monoculture*: the same systems, or systems that share components (e.g. datasets, models), are deployed by multiple decision-makers. While sharing offers advantages like amortizing effort, it also has risks. We introduce and formalize one such risk, *outcome homogenization*: the extent to which particular individuals or groups experience the same outcomes across different deployments. If the same individuals or groups exclusively experience undesirable outcomes, this may institutionalize systemic exclusion and reinscribe social hierarchy. We relate algorithmic monoculture and outcome homogenization by proposing the *component sharing hypothesis*: if algorithmic systems are increasingly built on the same data or models, then they will increasingly homogenize outcomes. We test this hypothesis on algorithmic fairness benchmarks, demonstrating that increased data-sharing reliably exacerbates homogenization and individual-level effects generally exceed group-level effects. Further, given the current regime in AI of foundation models, i.e., pretrained models that can be adapted to myriad downstream tasks, we test whether model-sharing homogenizes outcomes across tasks. We observe mixed results: we find that for both vision and language settings, the specific methods for adapting a foundation model significantly influence the degree of outcome homogenization. We also identify societal challenges that inhibit the measurement, diagnosis, and rectification of outcome homogenization in deployed machine learning systems.

## 1 Introduction

Machine learning is built on strong traditions of sharing: we share datasets (e.g. ImageNet), models (e.g. BERT), libraries (e.g. PyTorch), optimizers (e.g. Adam), evaluations (e.g. SuperGLUE) and much more. This ethos of sharing serves the field well: we are able to repeatedly capitalize on the effort required to build high-quality assets (e.g. ImageNet has supported thousands of researchers in computer vision), and improvements to these assets have sweeping benefits (e.g. BERT raised all boats in NLP). Yet does sharing also have risks? Could this central tenet of the field lead to undesirable outcomes?

---

[*]Corresponding author.

36th Conference on Neural Information Processing Systems (NeurIPS 2022).

We observe that certain forms of sharing can be reinterpreted as monoculture: Kleinberg and Raghavan [2021] define *algorithmic monoculture* as the state "in which many decision-makers all rely on the [exact] same algorithm." In parts of society where algorithmic systems are ubiquitous, we see trends towards such monoculture [Moore and Tambini, 2018, Engler, 2021]. Monocultures often pose serious risks: Kleinberg and Raghavan [2021] show monoculture is suboptimal for decision-makers when their decisions are interconnected, as when they compete to hire job candidates. In ML, our sharing practices often are more complex than sharing the entire algorithmic system: should we think of our practices of sharing assets in ML as monoculture and, if so, what harms should we worry about?

We investigate this question by proposing one potential risk we call *outcome homogenization*, i.e. the phenomenon of individuals (or groups) exclusively receiving negative outcomes from *all* ML models they interact with. For example, a job applicant may be rejected from every job they apply to due to the use of similar algorithmic resume screening systems at all companies. We view outcome homogenization as an important class of *systemic* harms that arise when we study social *systems*, i.e. harms that require observing how individuals are treated by many decision-makers.[2] In §2, we conceptually motivate outcome homogenization in the context of algorithmic hiring. In §3, we introduce the first mathematical formalism for outcome homogenization: we measure homogenization as the observed probability of systemic failure normalized by the base rate.

To link the practice of sharing in ML with the proposed harm of homogenization, we pose and test the *component sharing hypothesis*: algorithmic systems built using the same underlying components, such as training data and machine learning models, will tend to systematically fail for the same individuals or groups. We see component sharing as a specific form of algorithmic monoculture, which broadens the initial definition in Kleinberg and Raghavan [2021] from decision-makers deploying the *same* system to deploying *similar* systems in terms of how they are constructed. We investigate how two types of shared components — training data and foundation models — contribute to homogeneous outcomes.

In §4, we demonstrate that data-sharing reliably homogenizes outcomes for individuals and for racial groups, especially for small training datasets involving US census data. In §5, we discuss how the rise of foundation models [Bommasani et al., 2021], i.e. pretrained models that can be adapted to myriad downstream tasks, could yield unprecedented homogenization. Based on experiments with foundation models for vision (CLIP) and language (RoBERTa), to our surprise, we find the use of foundation models does not always exacerbate outcome homogenization. Instead, we find the specific mechanism for adapting the foundation model to the downstream task significantly influences homogenization: for example, linear probing consistently leads to more homogeneous outcomes than finetuning for both modalities. Through these experiments, it is clear that the relationship between sharing and homogenization is not fully explained by our hypothesis, but that there is some evidence that sharing homogenizes outcomes. To advance the study of homogenization in practice, where systemic harms are most consequential, we conclude by identifying key challenges for diagnosing, measuring, and rectifying homogenization in society (§6).

## 2 Outcome Homogenization in Resume Screening

To illustrate outcome homogenization and its potential causes (including algorithmic monoculture), we will use the example of algorithmic resume screening. Companies use resumes to screen job applicants, choosing which candidates to interview and which to reject. Maximum homogenization occurs when every company makes the same decision about each candidate, such that each lucky candidate is interviewed by all companies and each unlucky candidate by no companies. We say that the unlucky candidates who receive no interviews experience a *systemic failure*.[3]

**What factors might homogenize outcomes in human decision-making?** Even in the absence of the algorithms, we observe homogeneous outcomes in many setting. In hiring, historically, hiring

---

[2]In fact, outcome homogenization is a systemic harm that may arise even in the absence of algorithmic monoculture, though this work is restricted to settings where monoculture is present.

[3]A fundamental consideration is that the right or just outcomes in hiring are contested: the notions of merit and ground truth are much more subjective than, say, classifying images as dogs or cats. For this illustrative example, we do not delve into this, though we it makes explicit that some individuals can be justifiably rejected from all opportunities (e.g. those attempting to become lawyers without passing the bar exam). Hence, the interpretation of homogeneous outcomes will need to be contextual, and is likely to be value-laden in allocative contexts such as hiring, education, lending, and health.

managers at each company decided who to interview and often agreed in their decisions. This agreement can be attributed to multiple sources: first, if the needs of each company were identical, then managers at different companies may be incentivized to interview the same candidates, thereby homogenizing outcomes. Second, if hiring managers' choices are influenced by the same social biases, they will mistakenly reject the same people, thereby homogenizing their errors. Bias in resume screening is well-documented and remains significant [Jowell and Prescott-Clarke, 1970, Bertrand and Mullainathan, 2004, Kline et al., 2021, *inter alia*].

However, neither explanation implies that systemic failures are inevitable. Since companies have different needs and resumes are imperfect predictors of success in role, the "best" candidates will likely differ across companies. Further, bias is not uniform across companies: Kline et al. [2021] find that 21% of firms were responsible for 46% of the racial bias in interview decisions. Even if decisions are influenced by the same group-level biases, different companies may choose different individual members of the advantaged and disadvantaged groups. Variance in company needs, in prevalence of bias, and in individual hiring manager preferences all make it more likely that different resumes survive the screening stage at different companies, ensuring some diversity in resume screening outcomes.

**How do these dynamics change with the introduction of automated decision-making?** Most large companies now use automated resume screening software to parse resumes and decide which applicants advance. As a stylized example, if every company deploys the *same* deterministic system and has the same hiring criteria, then outcomes will be necessarily homogeneous: individuals will either receive interviews at every company or be rejected by all of them (i.e. systemic failure). This example is not far from reality: a few major vendors dominate the marketplace for algorithmic resume screening with 700 companies, including over 30% of Fortune 100 companies, relying on Hirevue [Hirevue, 2021]. Thiractice of different companies deploying the same system is defined as *algorithmic monoculture* by Kleinberg and Raghavan [2021].

More generally, different companies may instead deploy *similar*, but non-identical, systems. We expand the definition of algorithmic monoculture to encapsulate this broader setting, which is also alluded to in Kleinberg and Raghavan [2021]. Engler [2021] describe this as the reality for college enrollment management algorithms, writing "there are a relatively small number (between five and 10) of prominent vendors in the enrollment management algorithm market, . . . their process and analytics are markedly similar. Since their processes seem relatively consistent, the outcomes might be as well — potentially leading to consistently good results for students who match the historical expectations of colleges, and consistently poor results for students who don't".

**Component Sharing Hypothesis.** In this work, we study systems that are related in how they are constructed, akin to what is described by Engler [2021]. We pose the **component sharing hypothesis** that relates such algorithmic monoculture with outcome homogenization: *If deployed algorithmic systems share components, outcome homogenization will increase (i.e. there will be more systemic failures).* In this work, we empirically test this hypothesis for two prominent forms of component sharing: (i) the sharing of training data in training all deployed systems (§4) and (ii) the sharing of the same foundation model for building all deployed systems (§5).

## 3  Formalizing Outcome Homogenization

While prior work [Kleinberg and Raghavan, 2021, Creel and Hellman, 2022] alludes to outcome homogenization, here we provide the first mathematical formalism of outcome homogenization.[4] In line with our running example of resume screening, we formalize outcome homogenization for individuals in terms of *systemic failures* (i.e. every algorithmic system fails for an individual). We then generalize to the group setting, where groups are systemically excluded rather than individuals, with a discussion of how these metrics relate to established fairness, robustness, and accuracy metrics (§3.4).

### 3.1  Formalizing Outcome Homogenization for Individuals

**Notation.** Since we define outcome homogenization as a systemic phenomenon, we consider a social *system* $\{h^i\}_{i=1}^k$ where every individual $j$ interacts each deployed models $h^i$. As an example, an indi-

---

[4]The formal model of Kleinberg and Raghavan [2021] is related, but substantially distinct. Concretely, their formalism considers harms experienced by decision-makers, whereas we center decision-subjects.

vidual $j \in [N]$ submits features $x_j^i$ (e.g. their resume) as input to company $i \in [k]$ to receive an outcome $h^i(x_j^i) = \hat{y}_j^i$ (e.g. an interview). Let $D^i$ be the empirical distribution of inputs $x^i$ for company $i$.[5]

To define a notion of failure, let $I^i(x_j^i)$ indicate if $\hat{y}_j^i$ is a negative outcome, i.e. individual $j$ experiences a negative outcome from model $h^i$. The failure rate for model $h^i$ is

$$\text{FAIL}(h^i) \triangleq \underset{x^i \sim D^i}{\mathbb{E}} I^i(x^i) = \underset{x^i \sim D^i}{\Pr}\big[I^i(x^i) = 1\big]. \tag{1}$$

Experimentally, we consider classification errors as failures (i.e. $I^i(x^i) \triangleq \mathbb{I}\big[h^i(x^i) \neq y^i\big]$), but other negative outcomes (e.g. rejections from hiring or educational opportunities; $I^i(x^i) \triangleq \mathbb{I}\big[h^i(x^i) = -1\big]$) also can be studied under our framework.

**Systemic failures for individuals.** If an individual exclusively experiences failure, we say they experience *systemic failure*. The *observed rate of systemic failure* SYSTEMIC FAILURE$(h_1,...,h_k)$ is

$$\text{SYSTEMIC FAILURE}(h_1,...,h_k) \triangleq \underset{j}{\mathbb{E}}\left[\prod_i I^i(x_j^i)\right] = \underset{j}{\Pr}\big[I^1(x_j^1) = 1 \wedge \cdots \wedge I^k(x_j^k) = 1\big]. \tag{2}$$

**Homogenization metric for individuals.** SYSTEMIC FAILURE quantifies homogeneous outcomes, but is difficult to compare across systems with different underlying accuracies: SYSTEMIC FAILURE will in general be higher for less accurate systems independent of a *specific* tendency to pick (i.e. fail) on the same person. While we may sometimes want to combine accuracy and outcome homogenization into an overall measure of utility or social welfare, which SYSTEMIC FAILURE$(h_1,...,h_k)$ implicitly does, we focus on a *relative* measure of homogenization that disentangles accuracy from homogenization. In particular, we are interested in outcome homogenization even, and perhaps especially, in systems that are highly accurate.

As a result, we measure individual-level outcome homogenization for a social system $\{h^i\}_{i=1}^k$ by normalizing the *observed* rate of systemic failure by the *expected* rate of systemic failure.

$$H^{\textbf{individual}}(h^1,...,h^k) \triangleq \frac{\text{SYSTEMIC FAILURE}(h_1,...,h_k)}{\prod_i \text{FAIL}(h^i)} = \frac{\mathbb{E}_j\left[\prod_i I^i(x_j^i)\right]}{\prod_i\left[\mathbb{E}_j I^i(x_j^i)\right]} \tag{3}$$

This measure is the ratio between (i) the probability that an individual experiences systemic failure and (ii) the probability that randomly sampled outputs for each model are all failures. That is, the measure captures how the rate of systemic failure changes when we attend to the structure of individuals.

### 3.2 Formalizing Outcome Homogenization for Groups

In addition to individual-level homogenization, we also measure group-level homogenization. While our individual-level metric individualizes harm, complementing work on group-level biases, we may also want to identify the extent to which (possibly marginalized) social groups (e.g. Black women) are systemically excluded. Further, we often lack individual-level information (e.g. due to privacy concerns; see §6), or study algorithmic deployments that do not share individuals (e.g. hiring in different states).

**Notation.** For each input $x^i$, denote the associated group as $G(x^i) \in \mathcal{G}$. Group identity can correspond to the data producer (e.g. the age of a user querying a search engine) or the data subject (e.g. the race of an individual subject to face recognition). Let $D_g^i$ be the empirical distribution of inputs for group $g$ (i.e. $\{x^i \mid G(x^i) = g\}$). The *group failure rate* FAIL$_g(h^i)$ is

$$\text{FAIL}_g(h^i) \triangleq \underset{x^i \sim D_g^i}{\mathbb{E}} I^i(x^i). \tag{4}$$

**Homogenization metric for groups.** To measure group-level homogenization, we modify our individual-level metric with a weighted average over groups in place of a simple average over individuals.

---

[5]Note that our framework is general: we permit the deployed models to be for different tasks and for the individual's inputs to not be the same, though in our resume screening example all the models perform the same task and applicants often submit the same resume to different companies.

$$H_G^{\textbf{group}}(h^1,...,h^k) = \frac{\sum_g \left[ W(g)\prod_i \text{FAIL}_g(h^i) \right]}{\prod_i \text{FAIL}(h^i)} \qquad (5)$$

**Weights.** We consider three weighting schemes, specified by categorical probability distributions $W$ distributed over $\mathcal{G}$ (full definitions in §A.1):

**Average** ($H_{\textbf{avg}}$) $W$ weights each group proportional to its *frequency* across all deployments.

**Uniform** ($H_{\textbf{unif}}$) $W$ is the uniform distribution, so $W(g) = \frac{1}{|\mathcal{G}|}$.

**Worst** ($H_{\textbf{worst}}$) $W$ assigns weight 1 to the group $g_{\textbf{worst}}$ with the highest systemic failure rate and 0 to all other groups. This reduces the numerator to simply be the systemic failure rate for $g_{\textbf{worst}}$.

We introduce these weight functions to clarify that, much like having both individual-level metrics and group-level metrics, we may want to weight groups differently in different circumstances. For example, weighting by frequency may provide a useful overall measurement of homogenization but obscure systemic exclusion experienced by minority groups or specifically the worst-off group.

### 3.3 Understanding our metrics

As a ratio of probabilities, our metrics take values in $[0,\infty)$ where 0 indicates no systemic failures, 1 indicates the observed rate matches the expected rate, and values greater than 1 indicate some degree of outcome homogenization. In the individual setting, we assume each individual generates exactly one input per deployment, which may not hold in practice (e.g. people may submit multiple resumes or not apply to every company). We appropriately generalize our individual-level metric to address this in Appendix A. Further, in the group setting, we recover the individual-level metric using the **uniform** weighting (or the **average** weighting) if each individual's inputs are treated as belonging to their own group.

### 3.4 Relationship with other metrics

Since we introduce (several) metrics, we consider how they relate to metrics for related constructs (e.g. accuracy, fairness, robustness). This speaks to the convergent and divergent validity of our metrics [Campbell and Fiske, 1959, Messick, 1987, Jacobs and Wallach, 2021], i.e. whether they are adequately correlated with metrics of similar constructs and adequately uncorrelated with metrics of dissimilar constructs. Here, we discuss theoretical relationships, whereas in §5.2 we look at the empirical correlations.

**Accuracy.** When failure are errors, we design our metrics to minimize (anti-)correlation with accuracy. While not theoretically guaranteed, we empirically demonstrate this in Table 1. With that said, we expect there will be settings where the two are correlated: our goal is not to technically ensure no correlation, but to ensure that we do not neglect homogeneous outcomes in highly accurate systems (i.e. neglect the individuals who are systemically failed even when the overall picture may seem rosy).

**Fairness and Robustness.** Beyond accuracy, outcome homogenization is closely related to fairness and robustness. However, we emphasize that outcome homogenization is fundamentally about correlated outcomes for social *systems*, whereas almost all robustness or fairness metrics are defined for a single model. Recent work [Zhao and Chen, 2019, D'Amour et al., 2020, Wang et al., 2021] has initiated the study of fairness in multi-task learning, however these works focus on favorable overall trade-offs across tasks as opposed to systemic modes of failure. Conversely, our metrics cease to be interesting (e.g. $H^{\textbf{individual}}$ is always 1) in the single-model setting as systemic failures are single-model failures.

At a more fine-grained level, algorithmic fairness metrics [e.g. Dwork et al., 2012, Hardt et al., 2016] emphasize *discrepancies between individuals/groups*. In contrast, our metrics do not (explicitly) center these differences: we are interested in the observed rate of systemic failures (and whether this exceeds the expected rate). Performance differences are not *sufficient* for outcome homogenization: if the performance disparities for each deployment do not align across deployments, then outcomes may not be homogeneous. For robustness metrics, our metric $H_{\textbf{worst}}$ in the worst-case setting closely resembles the metrics studied in work on worst-group robustness [e.g. Sagawa* et al., 2020]. In particular, when there is only one deployment, our metric recovers the standard worst-group accuracy normalized by the overall accuracy.

## 3.5 Alternative metrics

In Appendix A, we more extensively discuss desiderata for our metric, alternatives we considered, and how we arrived at the metrics we present in the main paper. With that said, we also note conditions where we may instead favor alternatives, as well as connections to familiar quantities like the covariance, Pearson correlation, and (pointwise) mutual information in the binary setting ($k = 2$).

## 4 Data-Sharing Experiments

Having stated our mathematical formalism and metrics for outcome homogenization, we test if sharing training data leads to outcome homogenization. We consider widely used algorithmic fairness datasets [Fabris et al., 2022]: systemic failures for these specific datasets may not be of direct social consequence, but the datasets capture relevant social contexts where other forms of inequity have been documented.

**Data.** We work with two datasets: **German Credit** [**GC**; Dua and Graff, 2017], the third most widely used fairness dataset, and **ACS PUMS** [Ding et al., 2021], which was built to replace the most widely used fairness dataset, **UCI Adult**.[6] **GC** contains information on 1000 German contracts (e.g. credit history, credit amount, credit risk for the individual); following Wang et al. [2021], we consider two prediction tasks of (i) predicting if the individual receives a good or bad loan and (ii) predicting whether their credit amount exceeds 2000. **ACS PUMS** contains US Census survey data recording 286 features (e.g. self-reported race and sex, occupation, average hours worked per week) for 3.6 million individuals. Ding et al. [2021] construct several predictions tasks of which we use three: (i) predict if an individual is employed, (ii) predict an individual's income normalized by the poverty threshold, and (iii) predict if an individual has health insurance.

**Individuals and Groups.** For both datasets, we have individual-level information, hence we measure individual-level homogenization across models for each task. For **ACS PUMS**, we have self-identified race across 9 US Census categories (e.g. American Indian, Asian, Black/African American, White, two or more races), hence we measure group-level racial homogenization.

**Experimental Design.** To test if data-sharing influences outcome homogenization, we execute a controlled comparison by specifying two sampling protocols for the training data: **fixed** and **disjoint**. In general, articulating and determining what it precisely means for models to share data is complicated in natural settings: here we make the concept very precise. In the **fixed** setting, we sample $n$ points without replacement from the entire training dataset, which we use to train all of the $k$ task-specific models ($k = 2$ for **GC** and $k = 3$ for **ACS PUMS**). In the **disjoint** setting, we sample $kn$ points without replacement that we randomly partition across the $k$ task-specific models. In other words, in the **fixed** setting, the task-specific models *share the exact same training data inputs*, whereas in the **disjoint** setting, the task-specific models *share the same training distribution, but not the exact data*. We emphasize that this is a subtle difference between the setting, but it implies **fixed shares more** than **disjoint**.

Having specified the training data, we train models for each of the $k$ tasks (we consider all of logistic regression, SVMs, gradient-boosted trees, and small neural networks in Appendix C). To account for randomness, we report results averaged over 25 trials of the experiment (i.e. 5 samples of training data and 5 training runs per sample for every value of $n$ we consider).

**Results and Analysis.** In Figure 1 (**left**), we see clear evidence for our hypothesis: the **fixed** setting reliably shows more homogeneity than the **disjoint** setting. Overall, across all 3 datasets and 4 learning algorithms we consider, we find that **fixed** generally leads to more homogeneous outcomes than **disjoint**, which provides evidence towards our hypothesis: the use of the same training data leads to greater outcome homogenization than the use of different (but identically distributed) training data. However, this relationship is not perfect: we do see several instances where the degree of homogeneity is similar or even sometimes greater for the **disjoint** setting (e.g. regions of the **left** subplot of Figure 2). Therefore, data sharing alone does not fully characterize homogeneity (e.g. randomness in training and instability in the number of observed systemic failures are important to consider).

Further, the trends in the error rates for **fixed** and **disjoint** are near-identical (**right** subplot of Figure 1), as we would expect given the relationship between the sampling protocols. That is, only observing the accuracy is not enough in these settings for understanding the observed systemic failures, and provides evidence that our measures correctly identify discrepancies even when the underlying

---

[6]We include results for a third dataset, **LSAC** [Wightman et al., 1998], in Appendix C.

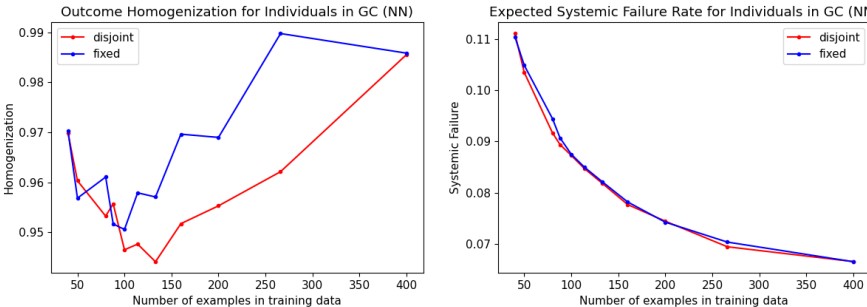

Figure 1: **Data-sharing leads to more homogeneous outcomes.** Data-sharing results for **GC** using neural network classifiers, which shows homogenization (**left**) and expected systemic failure rate (**right**) as a function of training dataset size ($x$-axis).

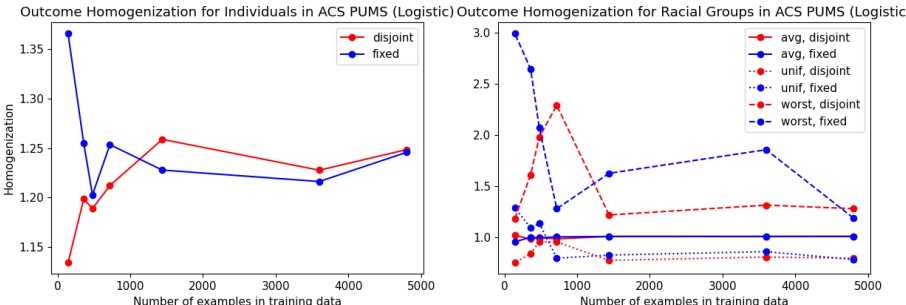

Figure 2: **Picking on the same person.** Data-sharing results for **ACS PUMS** using logistic regression: individual-level homogenization (**left**) exceeds group-level (**right**; **avg** and **unif**).

accuracies of each model are (approximately) the same. Finally, as a function of dataset size, we do not see consistent trends on the relationship between dataset scale and outcome homogenization, though we unsurprisingly see the **disjoint** and **fixed** settings converge for larger dataset sizes (since the discrepancies in the sampling become negligible due to convergence in measure).

**Picking on the same person.** For **ACS PUMS** in Figure 2, we contrast the individual-level and group-level homogenization. (Recall the **average** and **uniform** metrics are the group-level analogues of our individual-level metric.) Outcomes are consistently more homogeneous at the individual-group than for racial groups. In fact, group-level analysis show little homogenization (values near 1) and little change as a function of dataset size, whereas individual-level measurement exposes greater homogeneity and more variation (which is unsurprising since group-level quantities are more extensively aggregated). This has significant ramifications for many works on algorithmic fairness, which only consider social groups (e.g. race): these works may miss systemic failures for particular individuals that are obscured at the group-level [cf. Kearns et al., 2018, Hashimoto et al., 2018]. Even intersectional approaches may not suffice to surface these systemic failures, unless each intersectional group comprises a single individual.

## 5 Model-Sharing Experiments

Having found that data-sharing appears to exacerbate outcome homogenization, we now turn to model-sharing. Specifically, we test how sharing *foundation models* affects outcome homogenization. Bommasani et al. [2021] define foundation models as "models trained on broad data (generally using self-supervision at scale) that can be adapted to a wide range of downstream tasks". These models have had a sweeping impact on the AI research community, most notably in NLP, and are increasingly central to deploying ML at both startups (e.g. Hugging Face, Cohere, AI21) and established technology companies (e.g. Google, Microsoft, OpenAI). Sharing is endemic to the foundation model paradigm: to justify their immense resource requirements, models must be used repeatedly for costs to amortize favorably. In the extreme, if an entire domain like NLP comes to build almost all downstream systems on one or a few foundation models, then any biases or idiosyncrasies of these models that pervasively

manifest downstream could potentially yield unprecedented systemic failures and outcome homogenization [Bommasani et al., 2021, Fishman and Hancox-Li, 2022]. We see initial evidence for such algorithmic monoculture: BERT was downloaded 10 million times in the past month[7] alone and GPT-3 enables hundreds of deployed apps. Consequently, we believe it is especially timely to understand if, and to what extent, outcomes get homogenized as these models become entrenched as infrastructure.

## 5.1 Experiments

**Data.** To test how foundation models (e.g. CLIP, RoBERTa) influence homogenization, we run experiments for both vision and language data.[8] On the vision side, we work with the **CelebA** dataset [Liu et al., 2015] of celebrity faces paired with annotations for facial attributes. For each face image, given the associated attributes, we define two tasks (**Earrings**, **Necklace**) that involve predicting whether the individual is wearing the specific apparel item. Attribute prediction in CelebA has been studied previously in work on fairness and robustness [Sagawa* et al., 2020, Khani and Liang, 2021, Wang et al., 2021]. On the language side, we use four standard English text classification datasets following Gururangan et al. [2019]: **IMDB** [Maas et al., 2011], **AGNews** [Zhang et al., 2015], **Yahoo** [Chang et al., 2008], and **HateSpeech18** [de Gibert et al., 2018].

**Individuals and Groups.** Since the vision tasks are all based on CelebA, we have individual-level information. However, since the language tasks involve entirely different data (e.g. movie reviews vs. news articles), there is no (shared) individual-level information. At the group-level, for vision we use annotations for *hair color* and for whether the individual has a *beard*, whereas for language we automatically group inputs by *binary gender*.

**Experimental Design.** To test if model-sharing influences outcome homogenization, we contrast setting with differing degrees of model-sharing. In the vision experiments, we produce task-specific models for each task by either (i) training from **scratch** on CelebA data, (ii) linearly **probing** by fitting a linear classifier on features from the CLIP foundation model [Radford et al., 2021], or (iii) **finetuning** CLIP. To ensure meaningful comparisons, the models trained from scratch shared the same ViT architecture [Dosovitskiy et al., 2021] used in CLIP but with weights initialized randomly.

In the language experiments, we further hone in on the specific *adaptation method* used to adapt the foundation model (specifically RoBERTa-base [Liu et al., 2019]) to each task. We consider (i) linear **probing**, (ii) **finetuning**, and (iii) **BitFiT** [Ben Zaken et al., 2022], which is a recent *lightweight finetuning* method in NLP that involves freezing all the RoBERTa weights except the bias parameters which are updated as in finetuning. Consequently, BitFit is an intermediary between probing and finetuning, which has been shown to achieve similar accuracy as finetuning while updating very few of the pretrained parameters. For both vision and for language, all models are trained for the same number of epochs and we repeat each experiment for 5 random seeds per adaptation method.

**Hypotheses.** Much like data-sharing, model-sharing is graded and is not binary: different downstream systems can share varying degrees of underlying models. By design, our experimental design suggests a continuum in sharing: first, downstream system either can share a foundation model or not (**scratch**). Second, among methods that involve foundation models, all methods initialize the weights using the pretrained weights but differ in which parameters remain the same *after* adaptation is completed: **finetuning** changes all the parameters, **BitFit** only changes the bias parameters, and **probing** changes none of the parameters. As a result, overall, we can rank methods from most to least sharing as (i) **probing**, (ii) **BitFit**, (iii) **finetuning**, (iv) **scratch**, which leads us to predict the degree of homogenization will also follow this ranking under our component-sharing hypothesis.

**Results.** In Figure 3 (**left**), across all vision settings, we surprisingly find that **scratch** is the most homogeneous, i.e. more homogeneous than either approach involving shared foundation models. This is the opposite of what we hypothesized: we posit that this may indicate model sharing is not the key explanatory variable for outcome homogenization here, but instead it is a more complex form of data sharing. Specifically, we conjecture that since the **scratch** models are only trained on **CelebA** data, whereas the others also are trained on the much larger WebImageText via the CLIP foundation model, this may mean that the models based on CLIP are effectively regularized from learning idiosyncrasies of **CelebA** that the **scratch** models acquire. This may more generally suggest that a more correct hypothesis around data sharing should factor in the relationship (e.g. distribution shift) between the

---

[7]`https://huggingface.co/bert-base-uncased` as of October 2022.

[8]Full reproducibility details for vision are in §B.2; for language are in §B.3.

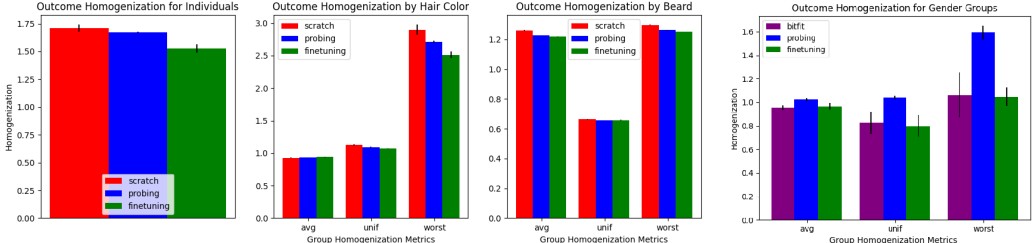

Figure 3: **Model-sharing does not reliably lead to more homogeneous outcomes.** Model-sharing results as a function of training/adaptation method for vision (**left three**) and language (**rightmost**). **Vision: scratch** is the most homogeneous, then **probing**, then **finetuning**. **Language: probing** is the most homogeneous; **finetuning** and **BitFit** are similarly homogeneous.

training data and the evaluation data for each model. Additionally, we find **probing** is consistently more homogeneous than **finetuning**, which aligns with our hypothesis. Finally, akin to the census results (§4), we once again find that outcome homogenization is significantly higher for individuals than for groups (comparing to $H_{\textbf{avg}}$ and $H_{\textbf{unif}}$).

In Figure 3 (**right**), across all language settings, we find the ordering of homogenization matches what our hypothesis predicts. Specifically, we find **BitFit** and **finetuning** achieve similar levels of homogeneity, even though **BitFit** updates $0.08\%$ of the parameters full finetuning does (i.e. the number of shared parameters for BitFit is more like probing than finetuning), suggesting the number of shared parameters is not the right lens for understanding model sharing. More broadly, these results do suggest parameter-sharing effects may contribute to outcome homogenization within the foundation model regime, but comparisons between foundation models and no foundation models may be more complex to explain.

### 5.2 Correlations between Metrics

Since we introduce several metrics, we measure the correlations between our metrics. Further, we measure correlations with accuracy (specifically, the expected rate of systemic failure) to test if homogenization is disentangled from accuracy. Since outcome homogenization is related to fairness, we also measure the correlation between our metrics and a standard group fairness metric. Fairness metrics are generally defined for a single model $h$, whereas we study entire systems $\{h^i\}_{i=1}^k$. We extend the unfairness definition used by Khani et al. [2019] as the variance in the systemic failure rates across groups.

$$\text{UNFAIRNESS}_G(h^1,...,h^k) \triangleq \underset{g}{\text{Var}}\left[\prod_i \text{FAIL}_g(h^i)\right] \tag{6}$$

**Results.** In Table 1, we report the pairwise correlation between metric pairs, based on the models we

| | **Vision** | | | | | **Language** | | | | |
|---|---|---|---|---|---|---|---|---|---|---|
| | $H_{\textbf{avg}}$ | $H_{\textbf{unif}}$ | $H_{\textbf{worst}}$ | Accuracy | Unfairness | $H_{\textbf{avg}}$ | $H_{\textbf{unif}}$ | $H_{\textbf{worst}}$ | Accuracy | Unfairness |
| $H_{\textbf{avg}}$ | - | *(0.87, 0.93)* | *(0.0, 0.96)* | (0.0, 0.09*) | (0.0, 0.8) | - | (0.22, -0.47) | (0.11, 0.56) | (0.06*, -0.22*) | (0.02, 0.09) |
| $H_{\textbf{unif}}$ | *(0.87, 0.93)* | - | *(0.0, 0.96)* | (0.0, -0.02) | (0.0, 0.74) | (0.22, -0.47) | - | *(0.63, -0.53)* | (0.0, 0.19*) | (0.0, -0.01) |
| $H_{\textbf{worst}}$ | *(0.0, 0.96)* | *(0.0, 0.96)* | - | (0.05, 0.1*) | *(1.0, 0.82)* | (0.11, 0.56) | *(0.63, -0.53)* | - | (0.02, 0.13) | (0.13, 0.47) |

Table 1: **Inter-metric correlations.** Correlations amidst our metrics as well as with other metrics reported as (Pearson $R^2$, Spearman $\rho$) with * significant at $p=0.05$ and *italics* significant at $p=0.001$.

trained in §5.1. These correlations are for 45 systems (3 methods × 3 groupings × 5 random seeds) of 2 models for vision and 15 systems of 4 models for language. For vision, our metrics are highly correlated with each other, whereas for language, $H_{\textbf{unif}}$ patterns quite differently (columns 1-3, 6-8). This is to be expected in that the vision groups (e.g. hair colors) all share similar frequencies, whereas the female group is significantly rarer in the language datasets. For both language and vision, we find that our metrics are generally not correlated, or perhaps weakly correlated, with accuracy as we intended (columns 4, 9). With respect to fairness, our worst-case metric $H_{\textbf{worst}}$ is strongly correlated for both modalities, but for the other two metrics we see no linear correlations and only monotone correlations for the vision experiments (columns 5, 10). This is in line with our broader expectations that fairness and outcome homogenization are indeed related (especially for the worst-performing group), but that given they are distinct theoretical constructs, they should not always be correlated [Campbell and Fiske, 1959].

### 5.3 Discussion

Across our experiments, we provide considerable evidence that sharing leads to homogeneous outcomes, but that it is incomplete explanation of homogeneity. This is particularly relevant when the findings in §4 and §5 are contrasted, given model-sharing in the foundation model regime indirectly implies immense data-sharing via the pretraining data (as mediated by the pretrained initialization). We emphasize that the regimes for these findings are quite different: low-dimensional tabular data with simple model families in data-sharing vs. high-dimensional images/text with large neural networks in model-sharing, so discrepancies in the findings may be attributable to these differences. More broadly, we believe a more complete explanation requires accounting for the data distributions and the associated distribution shifts (e.g. between pretraining and adaptation) at play. What we believe is clear, however, is that our findings provide an empirical basis to build on our conceptual arguments that sharing in machine learning can increase homogenization. This motivates investigation into real deployments of machine learning: for example, does sharing/monoculture lead to homogenization in algorithmic hiring.

## 6 Societal Considerations and Challenges

To situate our work in a broader social context, we identify and discuss core challenges in **diagnosing**, **measuring**, and **rectifying** outcome homogenization in real deployed systems.

**Diagnosis.** In our work, we posit monoculture yields homogenization: to follow this approach would require knowing which deployments rely on the same vendor, dataset, or foundation model (i.e. knowing where there is monoculture). Unfortunately, how algorithmic systems are constructed is often so opaque that identifying shared components is nigh impossible. However, if high homogenization were demonstrated, the measurement itself could justify provisions for increased transparency to identify the latent monoculture (i.e. the anti-causal direction). This provides a plausible mechanism for empowering auditors to be granted conditional access to otherwise inaccessible proprietary systems.

**Measurement.** Measuring homogenization only requires black box access, which is often achievable in practice [see Buolamwini and Gebru, 2018, Raji and Buolamwini, 2019, Metaxa et al., 2021]. However, identifying individual-level effects requires linking individual outcomes across deployments. Due to privacy constraints, linking individuals across different deployments may be challenging or impossible, which motivates group-level homogenization as more generally accessible (see §3.2).

**Rectification.** Even once outcome homogenization is identified, organizations may not be incentivized to reduce it. In fact, homogenization neither is attributable to any single entity nor can it always be addressed by unilateral action from a single organization. In the face of misaligned incentives and collective action problems, regulation, policy, or other compliance mechanisms may be required. Potential trade-offs between organization incentives and homogenization are further complicated if the harms of homogeneous outcomes take time to observe/accrue, but the benefits of, say, maximizing accuracy are immediate. More optimistically, Kleinberg and Raghavan [2021] show (under specific conditions) no trade-off exists between accuracy-maximizing policies and diversifying outcomes for societal benefit.

## 7 Limitations and Conclusion

We have introduced, formalized, and measured outcome homogenization as a systemic harm that may arise from practices of sharing in ML. Outcome homogenization is a new, understudied, and conceptually compelling topic: its definition, interpretation, statistical estimation, mitigation, and connections to monoculture remain poorly understood in spite of this work. We encourage future work to push in all of these directions. As to our measure, direct optimization may not lead to desirable outcomes, potentially even contributing to ethics-washing: its interpretation must be **contextual** since the implications of homogeneous outcomes heavily depend on broader societal context.

We believe homogenization is essential to holistically characterizing algorithmic harm, especially given growing monoculture (e.g. via foundation models). Without scrutiny, its harms may insidiously entrench. Consequently, we believe early intervention is necessary to prevent such harms in society.

**Reproducibility.** All code, data, and experiments are available on GitHub and CodaLab Worksheets.[9]

---

[9]https://worksheets.codalab.org/worksheets/0x807c29f8eb574d1fba8f429ec78b5d1b

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
