# OpenReview forum: "Picking on the Same Person: Does Algorithmic Monoculture lead to Outcome Homogenization?"
_NeurIPS.cc/2022/Conference — NeurIPS 2022 Accept_

### Official Review · Reviewer_h4Gz · 2022-06-22

**Rating:** 6
**Confidence:** 3
**Soundness:** 3 good
**Presentation:** 3 good
**Contribution:** 3 good

**Summary:**

The paper introduces a mathematical framework for characterizing outcome homogenization wherein homogenization is measured as the observed probability of systemic failure normalized by base rates. A conceptual motivation is also provided with the example of a algorithmic hiring. Experiments on US census data demonstrate that data sharing homogenizes outcomes for individuals and racial groups, this is more prominent for small training datasets.

**Questions:**

1. How would the type of learning algorithm influence homogenization? For e.g., studies have shown that supervised algorithms reinforce human bias especially in the context of scientific discoveries.

2. Related to the above question, would the extent of homogenization therefore be a function of the particular use case?

3.  Would the results be similar to tabular data?

4. How will dataset sparsity and measurement errors influence outcome homogenization?

5. Which of the factors mentioned in Section 2 are the most dominant factors in causing outcome homogenization - is there any intuitive or empirical evidence in support of this?

**Limitations:**

 see above

**Strengths And Weaknesses:**

Strengths:

1. The paper studies an important topic and can be valuable for further research in the context of fairness, accountability, and transparency of ML systems.

2. The claims made in the paper are backed by reasonable experimentation and exposition of the results

Weakness:

The paper would be much stronger with a more fine grained analysis across use cases and data modalities- see some pointers under questions below.

---

> ### Author Response · Authors · 2022-08-02
> **Response to Reviewer h4Gz**
>
> Thanks to reviewer h4Gz for their thorough review!
>
> We are glad they found the topic to be important and valuable, the claims to be backed by the experiments, and exposition to be appropriate overall. We took the core concerns to be (i) clarifying the role of learning algorithm/use case and (ii) addressing sparsity/measurement issues in our metrics — we have also run **additional experiments (more datasets, more models)** that clarify our findings, as well as **demonstrate how we address challenges in measurement.**
>
> **Questions 1-3 on learning algorithms, use cases, and tabular data.**
> To address Questions 1-3, we added new experiments with different learning algorithms (logistic regression, GBMs, SVMs, NNs) on tabular data (e.g. US Census). While the full results are provided in Appendix C, we find the **trends are qualitatively the same for data sharing**, but the absolute magnitude of homogenization changes. Similarly, across different datasets we find the use case doesn’t change the fact that sharing increases homogenization, but it does change by how much and the overall degree of homogenization varies substantially.
>
> **Question 4 on data sparsity and measurement errors.**
> To address data sparsity, **in Appendix A.3, we have given a variant of metric that addresses sparsity which is likely to arise in practice (e.g. individuals will not apply to every company)**. Further, as to other forms of sparsity, we do expect our metric to be brittle when there are few individuals for each application, as this means the failure rate for each application will be poorly estimated. Standard smoothing methods (e.g. Laplace smoothing) should help to alleviate this concern.
>
> **Question 5 on factors influencing homogenization in practice.**
> We believe there is a lot of desired agreement in preferences across firms, and that this may be further stratified by the nature of roles (e.g. roles like professor that require various qualifications may yield greater inherent agreement in preferences across employers when compared to roles like sales associate). For bias, we are less certain whether the group-level biases (e.g. higher hiring rates for men than women) will coincide with higher individual-level homogenization. We are actively conducting an in-progress audit of a resume screening vendor to clarify how important these factors are in practice, including as a function of role/occupation.

---

> ### Author Response · Authors · 2022-08-07
> **Any other questions or suggestions?**
>
> Thank you again for the comprehensive and useful review, and sorry to bother you. We wanted to check if you had any other questions or suggestions - we would love to get feedback to further improve our work!
>
> So far the review raised questions about the the data-sharing experiments as well as challenges in measurement. We hope the new experiments on 2 new datasets and 3 new methods address the former, and the discussion we have provided helps clarify the latter (in addition to Appendix A that delves into other generalizations of our measure to handle sparsity, etc.).

---

### Official Review · Reviewer_o96f · 2022-07-11

**Rating:** 6
**Confidence:** 4
**Soundness:** 2 fair
**Presentation:** 4 excellent
**Contribution:** 2 fair

**Summary:**

In this paper, the authors discuss how model and/or data sharing arising from a monoculture could lead to outcome homogenization, thus giving rise to several individual and group harms at a systemic level. Taking the example of resume screening, if a resume screening model misclassifies the resume of an otherwise well qualified individual and the same model is being utilized at every company, that individual will be experiencing systemic harm as they'll be rejected by every company. Similar case could be made for college admissions. The authors highlight that given recent advances in NLP, with large language models forming the basis of several AI applications, their sharing across applications and across companies could potentially be harmful to users for whom the underlying language model (that is being shared) does not perform accurately. The authors then extend this to group harms (systemic harms that could be faced by minority populations for instance). The research hypothesis that the authors then test is that *if algorithmic systems are increasingly built on the same data or models, then they will increasingly homogenize outcomes*.

The authors conduct experiments on data sharing and model sharing and find supporting evidence for their hypothesis in the data sharing regime, while finding mixed results in model sharing which suggest that the pretraining strategy used to train the underlying model likely plays a role in outcome homogenization.

While overall this is a sound paper, I have several questions. The paper is missing several inferential steps as one has to make some big (and potentially unrealistic) assumptions to arrive at a scenario where the same systems are being used by the same companies and/or are trained on the same data. Furthermore, the results on data sharing are not much different than the scenario where different datasets are used (when you reach closer to 10,000 datapoints). The practical viability of this problem is not established and I hope the authors will take time during the response period to address some of these issues.

-----------------------------------------------------------------------------------
I have updated my score following discussion with the authors but still have some concerns (reflected in my score) about the practical viability of the assumptions made in the experiments in the paper.

**Questions:**

- Did you try any experiments on data sharing with pretrained architectures? For example, you share the data you're using to finetune models but you're finetuning different pretrained models---that too is data sharing.
- Why did you run logistic regression five times on each sample? Correct me if I'm wrong but shouldn't the variance just be coming from different samples in case of a logistic regression model (this wouldn't have changed your plots as you're only plotting means but still curious).

**Limitations:**

I don't think the paper has done a good job at putting its limitations and assumptions up front.

**Strengths And Weaknesses:**

***Strengths***:
- The paper is well situated in prior work on monoculture.
- The experimental setups are well created and the discussion is particularly strong (except the assumptions).
- Limited results on data sharing suggest that there could be outcome homogenization in such scenarios and strikingly clear results on model sharing following certain forms of pretraining.

***Weaknesses***:
- The foundation of this problem is very fragile. It relies on a lot of assumptions being true, where it's unclear how true they might be. Counter example: Chase Bank would likely create its own model on it's own data (that it might merge with the Census data) to predict whether someone should be approved a loan or not. Citi Bank would create it's own. The corporations do not have an incentive to share models or data (or even rely on existing pretrained models when the normal practice in companies is to retrain their models frequently to account for new data) as they are competing against one another. Now let's look at resume screening. Take for example, Companis X and Y. Company X has different needs than Company Y despite both hiring for AI Research Scientists. Would Company X and Y want to use the same resume screening software trained on a generic dataset or have it at least be calibrated to their individual needs?
- Experiments on data sharing are limited to only Census data and it is unclear whether the trends hold beyond that dataset. In absence of additional results, a possible inference from these results could be that the trends you're showing are special to just the Census data.

***Minor comments***:
- The paper often makes broad generalizations when there are important qualifiers that need to be attached. For instance, Line 256 should be saying that data sharing appears to exacerbate outcome homogenization for small datasets...
- When providing examples in introduction (Adam, BERT, Pytorch, etc.), please include citations.
- Typo on Line 366: align aligns ---> align

---

> ### Author Response · Authors · 2022-08-02
> **Response to Reviewer o96f (1/3; monoculture in high-stakes domains and AI research)**
>
> Thanks to reviewer o96f for their thorough review! The main changes we made to address the concerns are:
>
> Extensive evidence that **monoculture arises in practice**
> 1. in high-stakes domains such as employment and education,
> 2. throughout AI research in various subareas such as vision and NLP,
> 3. throughout the foundation model ecosystem,
> 4. due to strong incentives for monoculture/sharing.
>
> Consequently, emphasize that we **believe monoculture/sharing is prevalent in many important settings, arguing instead that this core assumption is often true in practice rather than being fragile**. We also add **2 new sets of experiments (more datasets, more models)** to solidify data-sharing findings (Appendix C).
>
>
> > The foundation of this problem is very fragile. It relies on a lot of assumptions being true, where it's unclear how true they might be. … The corporations do not have an incentive to share models or data … as they are competing against one another. Now let's look at resume screening. … Company X has different needs than Company Y despite both hiring for AI Research Scientists. Would Company X and Y want to use the same resume screening software trained on a generic dataset or have it at least be calibrated to their individual needs?
>
> This is a very important concern: our paper centers the hypothesis that (more) monoculture/sharing yields more homogenization, hence we must verify that monoculture arises in practice for the harms of homogenization to be of significant concern.
>
> To address this concern, we build on the evidence we gave in Section 2 [lines 88-102] of monoculture in hiring and Section 5 [lines 262-270] of monoculture in foundation models.
> We provide evidence of the strong monoculture of Kleinberg and Raghavan (i.e. the exact same system is deployed by different decision-makers) and of the more general monoculture we study in our work (i.e. systems that share components are deployed by different decision-makers). To be clear, we do not claim monocultures are present in every setting, but we do claim that there are **strong pressures that incentivize monoculture** and that we already **observe data/model-sharing in many important and diverse settings**.
>
>
> **Monoculture in high-stakes domains.**
> - *Hiring.* At the level of models, more than 30% of the Fortune 100 companies use HireVue. To further understand how much sharing is going on, we are currently conducting an audit of a resume screening company. Even without model sharing, countless companies share the same data for hiring due to prominent data vendors such as Burning Glass Technologies that are known to be widely used throughout the hiring industry.
> - *Education.* Engler [1] provides a report on the state of algorithms used in the admissions process, reporting that “75% of colleges and universities” use analytics, which are generally provided by external vendors. While not all of these institutions use analytics to inform decision-making, many do in a “two-stop process” where algorithms are used in stage 1 and then supplement humans in stage 2. Engler writes “There are a relatively small number (between five and 10) of prominent vendors in the enrollment management algorithm market, … descriptions of their process and analytics are markedly similar. Since their processes seem relatively consistent, the outcomes might be as well—potentially leading to consistently good results for students who match the historical expectations of colleges, and consistently poor results for students who don’t.” **This reporting indicates both monoculture is quite salient in this sector, and posits the very harm we study in this work (as we cite in Section 2).**
>
> **Monoculture/sharing in AI research.**
> As we note in the Introduction, ML is built on strong traditions of sharing at many levels of abstraction (datasets, models, evaluations, optimizers, software libraries). For data, we note that for almost a decade, **most SOTA vision models are trained on ImageNet** and more recently **almost all SOTA NLP models are trained in some way on Wikipedia and CommonCrawl via pretrained components** (e.g. GloVE, BERT, GPT-3). For models, we see the same pretrained models in vision, NLP, speech for many years (e.g. ResNet-50, word2vec, GloVE, BERT, wav2vec 2.0); all of these works are **cited thousands of times because countless subsequent papers built models from these components**. In recent years, these components have also been used in other areas of AI (e.g. pretrained ResNets in robotics research for perception) and beyond (e.g. GloVE in various text analysis papers in the social sciences).

---

> > ### Author Response · Authors · 2022-08-02
> > **Response to Reviewer o96f (2/3; more evidence of monoculture for foundation models + incentives)**
> >
> > **Monoculture/sharing in the foundation model ecosystem.**
> > - *API providers.* Pioneered by OpenAI with GPT-3, many newer companies and startups (e.g. OpenAI, AI21, Cohere) provide API access to a foundation model as a commercial service. In this business model, **by construction, the foundation model is shared across many downstream clients** that adapt it for their respective use cases. In a press release in May 2021, OpenAI disclosed **over 300 downstream applications are built on GPT-3** and this has likely since grown. Examples of **companies deploying such GPT-based products** include Microsoft (e.g. GitHub CoPilot), Duolingo (to design questions that test user comprehension), Ought (for their AI research assistant, Elicit), as well as many copywriting and chatbot services (see https://leighmariebraswell.substack.com/p/overview-and-applications-of-large?sd=pf).
> > - *Established companies.* At established organizations (e.g. Google, Microsoft), users interact with a broad array of systems in their ecosystems (e.g. Google Search, Gmail, Google, Google Maps; Bing Search, Outlook, Microsoft Word). Within an organization, it is especially clear that sharing is incentivized, and given the social reach of these organizations, it is clear they have a profound impact on their users. Both Google and Microsoft have publicly disclosed through press releases they use foundation models in many of their flagship products (e.g. Google Search, Bing Search, Gmail, Google Docs, Microsoft Semantic Search, Microsoft Smart Replies). In fact, through private correspondence (that we will publish in forthcoming work), we have confirmed with leadership at Microsoft that they use **Turing-ULR language model** (https://www.microsoft.com/en-us/research/blog/microsoft-turing-universal-language-representation-model-t-ulrv2-tops-xtreme-leaderboard/) as **the foundation model in all of Bing Search** (for ranking, question answering, autosuggestion), in **Microsoft Word/Excel/PowerPoint/Outlook/Teams** (for Suggested edits/text predictions, semantic finding, summarization), **Semantic Search for enterprise customers**, etc., with the model being adapted (extensively) for each product.  That is, there is **very significant sharing of the same model** across many **high-profile and high-impact products at Microsoft**.
> >
> > **Incentives for Monoculture/Sharing.**
> > - *Data-sharing.* In domains where datasets are **scare or regulated**, different institutions often are forced to use the same data (e.g. biomedicine, law, education). Further, when there is **public/open data** that provides positive transfer, all organizations are incentivized to make use of it if their objective is to maximize accuracy. Similarly, when there are private data providers (as is standard in banking, quantitative finance, hiring), particular vendors often emerge as the dominant players by providing the **most efficient cost/benefit tradeoffs and numerous companies purchase their data**. Finally, if **curating** high-quality data is a key bottleneck, companies may outsource data acquisition to specialist vendors for data curation and cleaning.
> > - *Model-sharing.* In practice, foundation models are used at different companies because (i) they often **improve sample efficiency**, which is especially critical in data-limited domains and those where annotations are costly (e.g. law, medicine) as well as for smaller organizations that have access to less data, (ii) they often are much more **modular and convenient** to use due to complementary infrastructure like Hugging Face in NLP that makes using an LM very straightforward, and (iii) most fundamentally, they often allow organizations to **capitalize on the labor of other organizations for cheaper/zero cost** (e.g. BERT is free, GPT-3’s cost to a specific organization is likely far more favorable than producing a model a tenth of the size give GPT-3 costs millions of dollars to train beyond requiring numerous ML engineers, etc. that not all companies have).
> > - *Overall.* Overarching **rich-get-richer/Matthew effects** drive monoculture and monopolies in the economy. E.g. as HireVue is used by more companies for resume screening, they acquire more data to build even more accurate systems that incentivize broader adoption. I.e. standard economic dynamics yield positive feedback loops such that monopolies can induce monoculture. Similarly, **social norms and standards** can also build monoculture: the US Census is a fundamental data source because of its authoritative, high-quality, and high-status position, used by countless economists, sociologists, statisticians, and policymakers. In fact, we can turn to the origins of the term *monoculture* in agriculture: farmers who myopically maximized yield were incentivized to repeatedly plant the same crop, but ultimately this deteriorated the soil. Monoculture can be the **unintentional byproduct** of well-meaning optimization for individual objectives (often in the short term).

---

> > > ### Author Response · Authors · 2022-08-02
> > > **Response to Reviewer o96f (3/3; minor comments and references)**
> > >
> > > > Experiments on data sharing are limited to only Census data … results could be that the trends … are special to just the Census data.
> > >
> > > This is an important concern: we have added **two sets of new experiments** in the paper to address this (Appendix C currently). They confirm our **core findings generalize across datasets and methods**: see https://openreview.net/forum?id=-H6kKm4DVo&noteId=_e06JnB3uz for more details.
> > >
> > > > Minor comments
> > >
> > > We have addressed the typos and tightened the wording where the generalizations were too broad. Thanks!
> > >
> > > > Did you try any experiments on data sharing with pretrained architectures? For example, you share the data you're using to finetune models but you're finetuning different pretrained models---that too is data sharing.
> > >
> > > Thanks for the great suggestion - we are **running the NLP experiments** using BERT-base (same model size as the existing RoBERTa-base) and RoBERTa-large (larger model but trained on same data). We will post an update once we have these results (the original results took 180 hrs on a NVIDIA Titan XP as documented in Appendix B.3.2; BERT-base will take similar time and RoBERTa-large will take at least 3x longer).
> > >
> > > > Why did you run logistic regression five times on each sample? Correct me if I'm wrong but shouldn't the variance just be coming from different samples in case of a logistic regression model (this wouldn't have changed your plots as you're only plotting means but still curious).
> > >
> > > Yes, these is minimal/no variance due to model seed (see response to zUzT).
> > >
> > > **References.**
> > > - [1] Engler. Enrollment algorithms are contributing to the crises of higher education. The Brookings Institution 2021. https://www.brookings.
> > > edu/research/enrollment-algorithms-are-contributing-to-the-crises-of-higher-education/

---

> > > > ### Comment · Reviewer_o96f · 2022-08-07
> > > > **Clarification needed**
> > > >
> > > > I do agree there is a better case to be made of sharing in AI research and for foundational models under deployment but the other examples you share about monoculture and homogenization are very much in line with the hypothetical I shared. The way I currently read this paper, I see the motivation and the experiments (for model sharing) making the assumption that there is nothing different that two agents are employing in decision making, and all the decisions are a result of the automated toolkit that is being shared.
> > > >
> > > > But how often is that true and does introducing a slight of randomness into the process (say a human decision maker at the end who is only taking HireVue outputs into consideration as one of many) change the findings?
> > > >
> > > > Similarly, how often is it that when data sharing happens, it is just that dataset that is used in modeling and not the case that this particular dataset is merged with other data sources?
> > > >
> > > > There is a case to be made that this is also about access to technology: benign tools created using a GPT-3 API should still work across groups and if they don't then that's still an issue even if its not a high stakes setup.
> > > >
> > > > Would love to see the NLP experiments.

---

> > > > > ### Author Response · Authors · 2022-08-09
> > > > > **Response to Reviewer o96f (Clarification needed)**
> > > > >
> > > > > Thanks for the followup, we appreciate your engagement and are thankful for the opportunity to improve the work.
> > > > > In this case, your comment helped us better understand your concern, which we might not have fully understood the first time.
> > > > >
> > > > > We wanted to clarify two related but distinct matters:
> > > > > 1. Do we make sensible assumptions or are they fragile?
> > > > > 2. Is our setup/assumptions complete or are there other factors in play in practice?
> > > > >
> > > > > If the above distinction makes sense, we would be very happy to update the paper to better communicate these other factors based on your thoughts
> > > > >
> > > > > **Overall, we believe our assumptions are sensible/practical, but that there are other important factors at play in practice.**
> > > > > Our work is an initial attempt to study monoculture in a way motivated by practice. **Our generalized monoculture is much more common in practice than the very strict form of monoculture in Kleinberg and Raghavan**: Kleinberg and Raghavan require decisionmakers to deploy exactly the same system and each decision-maker to have the same task/objective, whereas we allow decision-makers to deploy different (but related) systems, to perform entirely different tasks (e.g. in our NLP experiments, the decision-makers would hypothetically be doing toxicity detection, sentiment analysis, and topic classification, which are pretty different tasks). As you describe, we think this broadening is quite important to better model reality.
> > > > >
> > > > >
> > > > > **Does monoculture happen in practice?**
> > > > > In AI research and with foundation models, we have greater visibility and it seems we agree the answer is yes. For the high-stakes settings we discuss (employment, education), unfortunately we do not have full visibility (in hiring, see “The Auditing Imperative for Automated Hiring Systems” by Ajunwa and “A Silicon Valley love triangle: Hiring algorithms, pseudo-science, and the quest for auditability” by Sloane et al.)
> > > > > Given what we do know, there definitely is some sharing. *Overall, our initial response aimed at showing such monoculture/sharing.*
> > > > >
> > > > > **Is monoculture/sharing the sole factor that influences variation in outcomes?**
> > > > > We think this may be at the heart of your concern: from what we know, we first want to note some cases in practice where people deploy the same system in the hiring setting. The key example hiring for a new role: for example, a company is hiring for sales associates for the first time. Necessarily the company has no internal data on hiring sales associates and if they use a third-party algorithmic hiring service, they will use the non-customized/base model provided by the service for that role. This is vaguely mentioned in this report from the automated hiring company pymetrics, though we have confirmed it in private correspondence as a concrete use case at pymetrics: https://workofthefuture.mit.edu/wp-content/uploads/2021/01/2021-Research-Brief-Polli-Kassir-Dolphin-Baker-Gabrieli.pdf.
> > > > >
> > > > > But with that example aside, we note there are lots of other important factors that (could and likely do) influence variation in the final decisions: in hiring, companies may have different preferences (as we describe in Section 2), companies usually combine human decision-making with algorithm decision-making (e.g. at pymetrics, they eliminate 70% of applicants and the remaining 30% are narrowed further by the company the threshold they interview), and companies also have differing fluidity/turnover that influence how the short-term interview outcomes connect to longer-term employment.
> > > > > **We definitely want to emphasize we are not capturing these other sources of variation and all the true complexity in practice; we want this work to be an initial step towards one important source of variation (sharing).**
> > > > >
> > > > > On data sharing, we agree there are several practical settings where the sharing is more complex due to merging, but important ones where it is not (e.g. biomedical or census data where the data is used as-is as the sole data - in these settings rather than merging with new data, if anything data is subselected). Also at newer companies, because they will lack internal user data, they may directly rely on what data vendors can provide.
> > > > >
> > > > > On **access to technology**, we think this is quite important. Our intent is definitely not to claim the high-stakes settings are more important, just that systemic exclusion in those cases is very clear in how much it can harm an individual. For example, an interesting setting to explore would be if **all commercial voice assistants fail for a user or all commercial face recognition fails to detect a specific person's face**; this would *complement work like GenderShades and Koenecke et al. that shows group-level biases in these systems*
> > > > >
> > > > > Finally, for the NLP experiments, they are unfortunately not yet done. We had to switch to A5000s for RoBERTa-large due to model size, so this led to an unexpected ~1.75x slowdown. We are sorry for this delay and will post them as soon as they are done.

---

> > > > > > ### Comment · Reviewer_o96f · 2022-08-09
> > > > > > **Appreciate the followup**
> > > > > >
> > > > > > I think this gets to the core of the issue. This discussion here is the nuance I was hoping to see. I would encourage you to include this framing in your paper. Thanks for the update on the NLP experiments. I’ll move my score to recommend acceptance.

---

> ### Author Response · Authors · 2022-08-07
> **Any other questions or suggestions?**
>
> Thank you again for the comprehensive and useful review, and sorry to bother you. We wanted to check if you had any other questions or suggestions - we would love to get feedback to further improve our work!
>
> So far the review raised questions about the validity of monoculture/its prevalence in practice as well as concerns about the generalizability of the data-sharing experiments.
> We hope our extensive evidence of monoculture (across various high-stakes settings, AI research, and the foundation model ecosystem) as well as incentives address the former, and the new experiments on 2 new datasets and 3 new methods address the latter.

---

### Official Review · Reviewer_zUzT · 2022-07-19

**Rating:** 4
**Confidence:** 4
**Soundness:** 2 fair
**Presentation:** 3 good
**Contribution:** 3 good

**Summary:**

The paper formalizes the idea of outcome homogenization, where a specific individual might receive negative outcomes from an array of different (but interconnected) models. The key contribution is to propose formal measures of homogenization and conduct analysis showing how homogenization relates to shared datasets and shared models used to carry out different tasks.

**Questions:**

Line 89: “not far from reality”. Does a lot of companies using the same provider/software indicate that the software outcomes on a given resume will be the same for all the companies?

Line 232: How are the 5 training runs different from each other? I understood how 5 samples of training data are different (drawn using different seeds), but could not understand the difference with training runs. Is it the model random seed? If yes, that should make no difference since the logistic regression solution is deterministic (owing to convexity).

Line 288: Is the whole transformer trained from scratch? How much time did it take? How did the accuracy compare to the fine tuned models?

Line 309: Do these results also consider different random seeds (there are some error bars in Fig 2 but maybe I missed how these are computed)? This might be quite important given the non-convex nature of these models.


**Limitations:**

In general the paper is quite up front about potential limitations. There are still many places where improvements can be made. For suggestions for improvements, please see the main review.

**Strengths And Weaknesses:**

The paper formalizes a very interesting and relevant notion of disadvantage which indeed may be prevalent given the widespread use of shared datasets and shared (foundational) models. While the prior works allude to the same notion (as the paper mentions in Section 3), a formalism and dedicated study has indeed been lacking. Formalizing homogenization opens up the doors for important analysis and insights, for instance the insight in Section 4 that even intersectional fairness metrics may not be able to detect systematic disadvantages. The paper is generally very well-written and easy to follow. Assumptions are stated rather clearly and formalism is followed by appropriate reasoning.

Unfortunately, the definitions and experiments seem quite preliminary. Definitions being preliminary is perhaps OK since the paper is only initiating research into a new area and later papers can study the topic in more detail. But drawing insights from the limited set of experiments presented here seems premature. For these reasons, I think the paper should be improved before it is ready for publication. Please see detailed comments for suggestions / questions:

1. The data sharing insights use just one single model (logistic regression) and dataset (ACS PUMS). How do we expect the insights to vary when we use other datasets or other models? Would the insights change if we use high parametrized models like NNs (where memorization is a concern)? Drawing insights from a single dataset seems insufficient.

2. It would also have been good to see some discussion into where the data sharing ends and model sharing starts. After all, two different versions of a foundational model still have a lot of training data in common (since the underlying model was pretrained on the same data). Given the fact that a large part of the training data is still shared, how do we reconcile the two sets of takeaways from Section 4 and 5?

3. I feel that the definition of homogenization might be too strict. Consider for example Eq. 2. Right now the quantity $\prod_i I^i$ is only 0 or 1 — indicating whether or not *all* systems misclassify the individual in question (even with 1 out of 100 systems going the other way, the takeaway would change completely). Why not use a soft version, for instance, fraction of systems that misclassify the individual?

4. I think the formulation in Section 3.2 can be simplified quite a bit. The weighted average of groups in Eq. 5 seems quite difficult to interpret. Why not simply measure which groups are being disadvantaged at a higher rate than others. In other words, why not measure the gap between the best and worst systematic group failure rates?

5. Line 119 and 120: First the paper mention $x_i^j$ but then the next line mentions $x_j^i$, causing a confusion in the terminology.

6. Looks like Figure 1 is quite sparse in terms of number of points. Given the large change that happens between $10^1$ and $10^3$, it might make sense to add more points in this region. Also, since the results anyway plateau quite quickly, why not consider a non-log scale? Finally, it would be great to add point markers indicating where the observed points are.

---

> ### Author Response · Authors · 2022-08-02
> **Response to Reviewer zUzT (1/n; new experiments)**
>
> Thanks to Reviewer zUzT for their thorough review! The main changes we made to address the concerns are:
> - **2 new sets of experiments (more datasets, more models)** to solidify data-sharing findings (Appendix C).
> - Extensive **grounding in social science theory on social hierarchy** to ground our strict condition for homogenization. We also provide **softer alternatives/generalizations** with a discussion of the **tradeoffs/circumstances** that make one preferable to the other.
>
>
> > The data sharing insights use just one single model (logistic regression) and dataset (ACS PUMS). … Drawing insights from a single dataset seems insufficient.
>
> This is an important concern: we have added **two sets of new experiments** in the paper to address this (Appendix C currently, with a pointer from the main paper).
>
> **Datasets.** We now include results on LSAC (predicting law school GPA and whether they pass the bar exam for US law students) and GC (predicting if a German individual has high credit and will receive a good loan). To identify these datasets, we include all multi-task datasets with human-centric outcomes in prior work on multi-task learning or fair ML [1, 2, 3].
>
> **Models/methods.** In addition to logistic regression, we include gradient-boosted trees, SVMs, and NNs.
>
> **Main Findings.** Qualitatively, we find **all of these evaluations reinforce our original core results**:
> - homogenization is consistently higher in the regime with greater data-sharing
> - homogenization is consistently higher at the individual level than the racial group level (in datasets where we have group information).
>
>
> > It would also have been good to see some discussion into where the data sharing ends and model sharing starts. …
>
> To directly address this, we **include discussion (Section 5.4) on this topic**, though we ultimately defer resolving this to future work as we see this as beyond the scope for the initial conceptual paper on this topic.
> We agree that the relationship between data-sharing (in the form of pre-training) vs. data-sharing in the distribution of interest is very interesting, and likely implies a more complete understanding of sharing will need to factor in distribution shift.
>
>
> > I feel that the definition of homogenization might be too strict. … Why not use a soft version, for instance, fraction of systems that misclassify the individual?
>
> We agree that the condition is strict, and brittle in the sense you note, but this is intentional: we believe the distinction of exclusive failure vs. not (e.g. rejected from all colleges, rejected from all jobs, no commercial product works for the individual) is quite fundamental. In the examples of hiring or education, this transition from 0 to 1 positive outcome has an **out-sized and profound impact on the individual** in question: it distinguishes whether they are employed or get into a university vs. not.
>
> To solidify the importance of this distinction, we point to extensive literature in the social sciences on structural inequality/social hierarchy, which explicitly identify this distinction as critical. In brief, this literature (under the heading of relational egalitarianism) explicits identifies **the harm of the same hierarchy leading to some individuals being entirely excluded**. **Consistent exclusion** from foundational social goods (like education, employment, credit) **risks establishing a social hierarchy of esteem or domination**. We develop on this in the next message.

---

> > ### Author Response · Authors · 2022-08-02
> > **Response to Reviewer zUzT (2/n; homogenization metric)**
> >
> > > I feel that the definition of homogenization might be too strict. … Why not use a soft version, for instance, fraction of systems that misclassify the individual?
> >
> > **Social Science Theory.** We follow Anderson [7] in taking a relational egalitarian approach. While individual organizations may establish rankings of candidates, and indeed we would expect that companies within a market sector will often agree on a hierarchy, the same hierarchy should not consistently dominate an entire sector or territory such that some people are entirely excluded from work [8, pg 74]. Anderson argues that “to be capable of functioning as an equal participant in a system of cooperative production requires ... access to the education needed to develop one’s talents, freedom of occupational choice, the right to make contracts and enter into cooperative agreements with others, the right to receive fair value for one’s labor, and recognition by others of one’s productive contributions” [7, pg 318]. If some people are consistently excluded from job interviews and therefore employment, they will not enjoy freedom of occupational choice; if they are excluded from higher education they will struggle to develop their talents. Not only do those excluded personally suffer from the establishment of the hierarchy, they also are unable to function as equal participants in society. Because employment, education, and credit are foundational social goods, consistent exclusion from them risks establishing a social hierarchy of esteem or domination. The hierarchy of esteem in turn damages the ability of the excluded to relate to others as equal democratic citizens [7].
> >
> > Further, the harm of exclusion from opportunities is of greater moral importance at scale since it is a threshold harm to autonomy. If autonomy is access to a sufficient range of sufficiently varied opportunities, then it is not a harm to be denied one opportunity, such as a job or a loan [9]. It is a harm, however, to be shut out of all (or a significant fraction of) opportunities. For this reason, in this work, we study the extreme case in which individuals are shut out of all opportunities.
> >
> > **Technical extensions of metric.** Returning to the technical metric, **in Appendix A.4, we do state analogous metrics for a softer loss and other forms of aggregation, rather than the all-or-nothing**, and discuss the **tradeoffs** (as well connections to well-studied quantities like the covariance and mutual information). Experimentally, our data generally only has a few tasks (e.g. 3 for Census, 2 for vision, 4 for NLP), but we agree softer variants are likely more apt for when k is large (e.g. someone applies to 30 places for faculty jobs).
> >
> > In terms of your suggestion of the fraction of systems, this may degenerate (e.g. if we replace the products with sums in the numerator/denominator, then they are the same by commutativity). More philosophically, we think there is something fundamentally different about person A getting 2 interviews and B 0 vs. both getting 1 that we want to emphasize.
> >
> > **TL;DR.** We believe the strict distinction is quite important with strong and deep connections to work on social hierarchy and structural inequity in the social sciences. With that said, softer formulations also have merits, especially when k is large, and we do state alternatives that are more suitable under these conditions.
> >
> > > I think the formulation in Section 3.2 can be simplified quite a bit. … why not measure the gap between the best and worst systematic group failure rates?
> >
> > We agree that measures like the one you describe are of interest, but they fundamentally measure notions of *fairness* (i.e. discrepancies in how groups are treated) rather than *homogenization* (i.e. are groups systemically excluded beyond what can be expected).
> > We clarify this in lines 185-201 (relationship with related concepts). The metric you state is quite close to the fairness metric we compared against of Khani et al. [10] in Equation 5; **we show this metric is weakly/uncorrelated with our metric in Table 1**.
> >
> > More broadly, we absolutely agree metrics like the one you describe are relevant in advancing the discourse on systemic harm.
> > Our objective in this work is to introduce and argue for why we should be measuring systemic harm; we encourage future work to then build on this to show the differential impact on some groups when contrasted with others.

---

> > > ### Author Response · Authors · 2022-08-02
> > > **Response to Reviewer zUzT (3/3; minor comments and references)**
> > >
> > > >Line 89: “not far from reality”. Does a lot of companies using the same provider/software indicate that the software outcomes on a given resume will be the same for all the companies?
> > >
> > > Using the same provider does not guarantee the same outcomes (e.g. some providers allow for customization). This observation precisely is **why we study the more general form of monoculture compared to the monoculture in Kleinberg and Raghavan where systems must be identical: customization is a form of the monoculture we study**, where the base system is the shared component and the per-company customization is the adaptation process, because it is unclear whether the outcomes will be the same from all companies.
> > >
> > > In practice, the exact practices in the resume-screening industry are not well-documented; we are actively conducting an audit that builds on this work to clarify this. Our initial findings are that companies do customize and that companies request different thresholds for the number of candidates to be interviewed. That is, we observe the more general form of monoculture we study in our work in practice, though we don’t always see the stronger form of exact sharing studied in Kleinberg and Raghavan.
> > >
> > > > Line 232: How are the 5 training runs different from each other? … Is it the model random seed? … should make no difference .. due to convexity.
> > >
> > > Yes, these are model random seeds and, indeed, by convexity there should be no variation. Nonetheless we included it to ensure the variation (in the solution, but also accuracy/homogenization measured at the solution) yielded from the solver in `sk-learn` was marginal, which we did find to be the case (though not perfectly zero).
> > >
> > > > Line 288: Is the whole transformer trained from scratch? How much time did it take? How did the accuracy compare to the fine tuned models?
> > >
> > > **All details are provided in Appendix B.2.2:**
> > > The whole transformer is trained from **scratch** in the scratch setting, o/w it is initialized with the CLIP weights.
> > > In aggregate, all runs took **1000 hours on a NVIDIA Titan XP** as we optimize hyperparams independently for each setting and do 5 random seeds per settings
> > > The average accuracy (over the 5 runs) for the Earrings task is **84.7 for scratch, 90.1 for probing, and 92.6 for fine-tuning** (i.e. using the foundation model considerably improves accuracy given fixed computation for building the downstream system). The **expected systemic failure rate of scratch is 0.023, for probing is 0.014, and for finetuning is 0.010.**
> > >
> > > To validate our implementation, we verified accuracies matched prior work [11]
> > >
> > > > Line 309: Do these results also consider different random seeds … This might be quite important given the non-convex nature of these models.
> > >
> > > Yes, we thought sensitivity to randomness was quite important to check for, so all results and error bars come from 5 runs (see line 299). Our new results on data-sharing also help better clarify this in a more controlled setting across different model families.
> > >
> > > > Comments on notations and figures.
> > >
> > > We corrected the typo and improved the figure visualization; thanks for the guidance!
> > >
> > >
> > > **References.**
> > > - [1] Wang et al. Understanding and Improving Fairness-Accuracy Trade-offs in Multi-Task Learning. KDD 2021.
> > > - [2] Zhang and Yang. A Survey on Multi-Task Learning. IEEE Transactions on Knowledge and Data Engineering 2017.
> > > - [3] Quy et al. A survey on datasets for fairness-aware machine learning. WIREs Data Mining and Knowledge Discovery 2021.
> > > - [4] Breiman. Statistical Modeling: The Two Cultures. Statistical Sciences 2001.
> > > - [5] Semenova et al. On the Existence of Simpler Machine Learning Models. FAccT 2022.
> > > - [6] Marx et al. Predictive Multiplicity in Classification. ICML 2020.
> > > - [7] Anderson. Liberty, Equality, and Private Government. University of Utah Press 2016.
> > > - [8] Anderson. What is the Point of Equality? Ethics 1999.
> > > - [9] Raz. The Morality of Freedom. Oxford University Press 1988.
> > > - [10] Khani and Liang. Feature noise induces loss discrepancy across groups. ICML 2020.
> > > - [11] Sagawa et al. Distributionally Robust Neural Networks. ICLR 2020.

---

> ### Author Response · Authors · 2022-08-07
> **Any other questions or suggestions?**
>
> Thank you again for the comprehensive and useful review, and sorry to bother you. We wanted to check if you had any other questions or suggestions - we would love to get feedback to further improve our work!
>
> So far the review raised questions about the validity/generalizability of the data-sharing experiments and the strict nature of our metric.
> We hope our response (Appendix C: additional experiments with 2 new datasets, 3 new methods that confirm our initial findings) has addressed the former, and our grounding in social science theory and alternative softer metrics has addressed the latter.

---

### Meta-Review · Area_Chair_uU6J · 2022-09-01

**Recommendation:** Accept
**Confidence:** Certain

**Metareview:**

This paper has the potential to catalyse a new and important line of research within algorithmic fairness. It does that (in a simple, yet interesting way) by identifying that the key to the concerns about algorithmic monoculture lies in homogeneous outcomes and by defining intuitively justified metrics of homogeneous outcomes at individual and group-levels. It then characterises the potential outcome homogenisation in models trained using the same/similar datasets and in models fine-tuned using the same underlying foundational models (a trend that is increasing, of late). To be fair, the reviewers rightly criticised for a somewhat limited set of experiments and the conclusions drawn from them do appear rather preliminary. However, this can be partly justified as the paper is exploring a previously unexplored territory and the potential benefits of accepting the paper and the follow-on work that it will very likely trigger far outweigh the potential risks of not accepting the paper (after all, the experiments are rigorous, even if limited / preliminary and the authors draw appropriate conclusions).

**Award:**

No

---

### Decision · Program_Chairs · 2022-09-14

Accept